# Analysis of Noise-Detection Characteristics of Electric Field Coupling in Quartz Flexible Accelerometer

**DOI:** 10.3390/mi14030535

**Published:** 2023-02-25

**Authors:** Zhigang Zhang, Dongxue Zhao, Huiyong He, Lijun Tang, Qian He

**Affiliations:** 1School of Physics and Electronic Science, Changsha University of Science and Technology, Changsha 410114, China; 2Hunan Provincial Key Laboratory of Flexible Electronic Materials Genome Engineering, Changsha 410114, China; 3Key Laboratory of Electromagnetic Environment Monitoring and Modeling in Hunan Province, Changsha 410114, China

**Keywords:** quartz flexible accelerometer, electric field coupling noise, lock-in amplifier circuit

## Abstract

The internal electric field coupling noise of a quartz flexible accelerometer (QFA) restricts the improvement of the measurement accuracy of the accelerometer. In this paper, the internal electric field coupling mechanism of a QFA is studied, an electric field coupling detection noise model of the accelerometer is established, the distributed capacitance among the components of the QFA is simulated, the structure of the detection noise transfer system of different carrier modulation differential capacitance detection circuits is analyzed, and the influence of each transfer chain on the detection noise is discussed. The simulation results of electric field coupling detection noise show that the average value of detection noise can reach 41.7 μg, which is close to the effective resolution of the QFA, 50 μg. This confirms that electric field coupling detection noise is a non-negligible factor affecting the measurement accuracy of the accelerometer. A method of adding a high-pass filter to the front of the phase-shifting circuit is presented to suppress the noise of electric field coupling detection. This method attenuates the average value of the detected noise by about 78 dB, and reduces the average value of the detected noise to less than 0.1 μg, which provides a new approach and direction for effectively breaking through the performance of the QFA.

## 1. Introduction

The accuracy of quartz flexible accelerometers is an important factor that affects the navigation performance of inertial navigation systems. According to public data, high-precision quartz flexible accelerometers such as QA3000-30 [1], AI-Q-2010 [2], and A600 [3] have been widely used in inertial navigation systems. The performances of these three QFAs are shown in Table 1. Among them, the QA3000-30 accelerometer developed by Honeywell in the 1990s has an accuracy of 1 μg, making it the most accurate and reliable QFA currently available. However, after more than 30 years of development, the effective accuracy of the QFA remains around 1 μg without a significant breakthrough [4]. The stagnation of the performance development of the QFA has gradually become the obstacle to overcome in improving the navigation performance of inertial navigation systems.

The noise in a QFA system mainly includes the noise caused by the change in external temperature and other environmental parameters, the mechanical thermal noise, the resonance noise [5], and the coupling noise [6] caused by the operation of the differential capacitance detection circuit and torquer drive circuit in the system. The impact of this noise on the performance of the accelerometer cannot be ignored. In recent years, there have been many studies on the effect of noise caused by temperature change on the performance of accelerometers [7,8,9,10,11]. The newly researched temperature compensation method maintains the stability of the cold start output at about 30 μg [12], and increases the stability of 1 g acceleration output to 14.6 μg [13], which significantly improves the temperature stability of the QFA.

In order to improve the accuracy of the QFA, the internal noise of the QFA system has attracted much attention from researchers, and a lot of meaningful work has been carried out on the differential capacitance detection circuit [14,15], the analysis of the noise characteristics of the drive circuit [16,17], and the design and optimization of the circuit structure. Huang et al. established a noise model of the readout circuit, analyzed the noise of the charge-discharge detection circuit and its equivalent acceleration formula, and found that the noise of the charge-discharge method is about 1 μg [18]. Ran et al. proposed a digital closed-loop QFA signal-detection method based on the AC bridge, and Chen et al. designed a differential capacitance detection method based on the capacitor bridge [19]. Both methods achieved a resolution for the detection circuit of 0.0018 pF, which matches the measurement of 1 μg resolution acceleration [20]. In these studies, the researchers only focused on the noise optimization of a single detection circuit or driving circuit, and did not pay attention to the coupling relationship between them. When a QFA is working, the driving signal in the torquer coil changes rapidly, and the electromagnetic radiation generated is coupled to the differential capacitance plate, which is introduced into the differential capacitance detection circuit to cause interference. The existence of electromagnetic coupling noise affects the performance of the QFA [21]. In this research, we studied the internal electric field coupling mechanism of a QFA, established an accelerometer electric field coupling detection noise model, analyzed the characteristics of the electric field coupling detection noise, and explored the influencing factors and suppression methods of the electric field coupling detection noise.

## 2. Analysis of Electric Field Coupling in QFA

### 2.1. Mechanism of Electric Field Coupling in QFA

A QFA is mainly composed of a quartz pendulum reed component, differential capacitance sensor, torquer, shell, etc. [22], and its structure is shown in Figure 1a. The quartz pendulum assembly, differential capacitance sensor, torquer, and drive circuit constitute the closed-loop system shown in Figure 1b [23]. The transfer function H(s) of the closed-loop system can be expressed as:(1)$$H\left(s\right)=\frac{I_{T}}{a_{i}}=\frac{K_{B}\theta \left(s\right)K_{s}K_{a}M\left(s\right)K_{I}}{1+\theta \left(s\right)K_{s}K_{a}M\left(s\right)K_{I}K_{T}}$$
where $K_{B}$ is the pendulum property of the pendulum reed component, and θ(s), $K_{s}$, $K_{a}$, M(s), $K_{I}$, and $K_{T}$ are the transfer functions of the pendulum reed component, differential capacitance sensor, differential capacitance detection circuit, torquer control system, torquer driving circuit, and torquer, respectively. The pendulum reed transfer function θ(s) of QFA is:(2)$$\theta \left(s\right)=\frac{1}{JS^{2}+CS+K}$$
where *J* is the rotational inertia, *C* is the damping factor, and *K* is the elastic factor. In the closed-loop system, the input acceleration $a_{i}$ can be obtained by measuring the drive current $I_{T}$, and its direction depends on the drive-current direction.

In the QFA, the shell, yoke iron, torquer coil, differential capacitor plate, and other components are made of metal materials. These components are independent of each other and do not make contact with each other. When the accelerometer works, there is a potential difference in electricity between different components, which forms the distributed capacitance shown in Figure 2a. $C_{d1}$, $C_{d2}$, and $C_{d3}$ are the distributed capacitances between the torquer coil and the plates of the differential capacitor; $C_{bb}$ is the distributed capacitance between the top and bottom plates of the differential capacitor; $C_{e1}$, $C_{e2}$, and $C_{e3}$ are the distributed capacitances between the three plates of differential capacitance and the accelerometer shell; and $C_{e4}$ is the distributed capacitance between the accelerometer shell and the torquer coil.

The distributed capacitance couples the torquer drive current $I_{T}$ to the differential capacitance detection process to form the detection noise $V_{n}$. Based on this, the equivalent circuit model of electric field coupling detection noise inside the accelerometer is established, as shown in Figure 2b. The controlled current source $I_{T}$ is the current output of the torquer drive circuit; $L_{T}$ and $R_{T}$ are the resistance and inductance of the torquer coil, respectively; and $R_{CS}$ is the current sampling resistance.

### 2.2. Simulation of Distributed Capacitance in QFA

When analyzing the noise characteristics of accelerometer electric field coupling detection, it is necessary to determine the value and variation law of each distributed capacitance. The value of the distributed capacitance in the QFA mainly depends on the directly opposite area and distance of each component. Only the pendulum reed in each component will move with the change in input acceleration, which will change the area and distance between the components. Therefore, it is only necessary to study the relationship between the distributed capacitance and the displacement of the pendulum reed. In this study, the finite-element simulation method was used for analysis and calculation.

The simulation model of the QFA was established according to the assembly structure and materials of a typical QFA. The main performance of the typical QFA is shown in Table 2.

The geometric dimensions of the main structures are shown in Table 3.

The material type and relative permittivity of each component in the simulation model are shown in Table 4.

Voltage excitation was applied to each component according to the actual working condition of the accelerometer head. The voltage excitation settings are shown in Table 5.

A negative displacement with a step length of 0.5 μm was applied to the quartz pendulum component along the sensitive axis, and the variation law of differential capacitance and distributed capacitance of the pendulum component was analyzed from the balance position to the deviation range of 19.00 μm. The accuracy of the simulation calculation was set to 1%, which considered both simulation efficiency and data accuracy.

The simulation results of differential capacitance and distributed capacitance are shown in Figure 3. The differential capacitance $C_{s1}$ decreases monotonically and $C_{s2}$ increases monotonically with the increase in pendulum reed displacement. When the movement of the pendulum reed is very small, the decrease in $C_{s1}$ is approximately equal to the increase in $C_{s2}$, and can be approximately expressed as:(3)$$\{\begin{array}{c} C_{s1}=C_{0}-\Delta C\\ C_{s2}=C_{0}+\Delta C \end{array}$$
where ΔC is the change value of the differential capacitance, and $C_{0}$ is the initial value of the differential capacitance when the pendulum reed is in the balance position, $C_{0}$ = 40 pF. Equation (3) conforms to the characteristics of the QFA differential capacitance sensor [24]. $C_{d1}$, $C_{d2}$, and $C_{d3}$ increased or decreased with the increase in pendulum displacement, but the variation was only ±0.05 pF; $C_{e1}$ and $C_{e2}$ have no obvious change trend with the increase in pendulum displacement; $C_{e3}$ and $C_{e4}$ have a decreasing trend, but the change is very small; $C_{bb}$ has a decreasing trend with the increase in pendulum displacement, and the variation is 0.03 pF.

The simulation results show that, within a certain error range, the value of the distributed capacitance is independent of the movement of the pendulum reed, which can be explained by the structure and working principles of the QFA. When the pendulum reed moves, the torquer coil wound on the coil frame will not move relative to the pendulum reed because it is bonded to both sides of it. The distributed capacitance $C_{d1}$ and $C_{d2}$ between the torquer coil and the top and bottom plates of the differential capacitor are independent of the movement of the pendulum reed. When the torquer coil moves with the pendulum, it is always between the upper and lower yoke iron. When the area of one side decreases, the area of the opposite side increases, and the two areas increase and decrease equally. The distributed capacitance $C_{d3}$ between the torquer coil and yoke iron remains unchanged. The metal coating on both sides of the pendulum reed is always static, and the distributed capacitance $C_{bb}$ between the top and bottom plates of the differential capacitor is fixed. The yoke iron and the shell are fixed, and the distributed capacitance $C_{e1}$ is fixed. In addition, the upper and lower yoke iron and the bellyband are welded using a laser to form an almost closed space, which produces an electrostatic shielding effect. The distributed capacitances $C_{e2}$, $C_{e3}$, and $C_{e4}$ between the components inside the enclosed space and the shell are very small. Therefore, when the electric field coupling detection noise equivalent circuit model is used to analyze the detection noise, the very small distributed capacitances $C_{e2}$, $C_{e3}$, and $C_{e4}$ can be ignored. The distributed capacitance values are shown in Table 6 and can be regarded as fixed values.

## 3. Analysis of Detection Noise Transfer System Structure

According to the equivalent circuit model of electric field coupling detection noise, the noise is related to the type of differential capacitance detection circuit. At present, the common differential capacitance detection circuits mainly include capacitance divider type, switch capacitance integral type, ring diode type and carrier modulation type detection circuits. Among them, the carrier modulation detection circuit is widely used in QFAs. The carrier modulation detection circuits are divided into single-channel carrier modulation (SCM) and dual-channel carrier modulation (DCM) detection circuits. In addition, the shell has two connection modes: grounded and float. Shell grounding is also a common electromagnetic shielding method. The float shell is designed to prevent the internal components from being destroyed in use. Different types of differential capacitance detection circuits and the connection mode of the accelerometer shell constitute multiple combinations of noise transfer paths.

### 3.1. Detection Noise Transfer System Structure of SCM Detection Circuit

When an SCM detection circuit is used to detect differential capacitance, the structure of the detection circuit is as shown in Figure 4a. The single-channel high-frequency sinusoidal carrier signal $V_{s}$ is input to the differential capacitance fixed plate, and the signals of the two moving plates pass through the charge amplification circuit and the differential amplification circuit to obtain the modulated signal $V_{a}$, whose amplitude reflects the variation in the differential capacitance ΔC.

The modulated signal $V_{a}$ can be expressed as:(4)$$V_{a}=\frac{C_{s2}-C_{s1}}{\frac{1}{j\omega R_{f}}+C_{f}}\frac{R_{2}}{R_{1}}V_{s}=\frac{2\Delta CA_{diif}}{\frac{1}{j\omega R_{f}}+C_{f}}V_{s}$$
where $A_{diif}=\frac{R_{2}}{R_{1}}$ is the amplification factor of the differential amplifier circuit. Generally, the value of $R_{f}$ ranges from dozens to hundreds of MΩ; thus, Equation (4) can be simplified as:(5)$$V_{a}=\frac{2A_{diif}}{C_{f}}\Delta CV_{s}$$

It is assumed that sin carrier signal $V_{s}=A_{s}\sin\left(\omega_{s}t\right)$ and differential capacitance variation $\Delta C=A_{c}\sin\left(\omega_{c}t\right)$, where $A_{s}$ and $A_{c}$ are amplitude $V_{s}$ and ΔC, respectively; $\omega_{s}$ is the frequency of carrier signal $V_{s}$, usually from tens of kHz to tens of MHz; and $\omega_{c}$ is the effective detectable frequency of ΔC, which is related to the bandwidth of the accelerometer closed-loop system, generally less than 1 kHz. We take the time-domain representation of $V_{s}$ and ΔC into Equation (5) to obtain:(6)$$V_{a}=-\frac{A_{diif}}{C_{f}}A_{c}A_{s}\left[\cos\left(\left(\omega_{s}+\omega_{c}\right)t\right)-\cos\left(\left(\omega_{s}-\omega_{c}\right)t\right)\right]$$

The modulated signal $V_{a}$ is mixed with the phase-adjusted carrier signal $V_{s+\varphi }=A_{s}\sin\left(\omega_{s}t+\varphi \right)$ through the mixer, and the mixing output signal $V_{b}$ can be expressed as:(7)$$V_{b}=-\frac{A_{diif}}{2C_{f}}A_{c}A_{s}^{2}\left[\sin\left(\left(2\omega_{s}+\omega_{c}\right)t+\varphi \right)-\sin\left(\omega_{c}t-\varphi \right)\right]$$

There are high-frequency signals of frequency $2\omega_{s}+\omega_{c}$ and low-frequency signals of the same frequency ΔC in $V_{b}$. When $V_{b}$ passes through a low-pass filter with a cut-off frequency of $\omega_{c}$, the high-frequency signals in $V_{b}$ are filtered out, and only the low-frequency signals in the same frequency as ΔC are retained. The filter output signal $V_{LPF}$ can be expressed as:(8)$$V_{LPF}=\frac{A_{diif}A_{s}^{2}}{2C_{f}}A_{c}\sin\left(\omega_{c}t-\varphi \right)$$

After $V_{LPF}$ adjusts the gain through the amplifier, the output voltage $V_{C}$ of the SCM detection circuit is obtained:(9)$$V_{C}=\frac{A_{diif}A_{Gain}A_{s}^{2}}{2C_{f}}\Delta C$$
where $A_{Gain}$ is the amplification factor of the amplifier circuit. According to Equation (9), the output voltage $V_{C}$ of the SCM detection circuit is directly proportional to the variation of differential capacitance ΔC.

According to the equivalent circuit model of electric field coupling detection noise and the simulation results of distributed capacitance, the detection noise transfer path formed by the SCM detection circuit is as shown in Figure 4b. After the voltage $V_{T}$ on the torquer coil passes through the distributed capacitance network, it is loaded onto the three plates of the differential capacitor, and then passed into the charge amplification circuit and the differential amplification circuit to form a pseudo-modulated signal $V_{m-s}$. The signal loaded on the fixed plate is also passed into the phase-shift circuit as a pseudo-carrier signal $V_{p-s}$. After mixing $V_{p-s}$ with $V_{m-s}$, the electric field coupling detection noise $V_{n-s}$ is formed through a low-pass filter and gain-adjustment circuit. In the SCM detection noise transfer path, $V_{T}$ forms a common-mode signal when it is transmitted to the detection circuit through $C_{d1}$ and $C_{d3}$, which is cancelled when it goes through differential amplification, and does not affect the differential capacitance detection. $C_{bb}$ does not divide voltage or affect differential capacitance detection. We remove $C_{d1}$, $C_{d3}$, and $C_{bb}$ to obtain a simplified detection noise transfer path of the SCM detection circuit, as shown in Figure 4c. In the transfer path of the detection noise of the SCM detection circuit, the distributed capacitance $C_{d2}$ between the torquer coil and the yoke iron and $C_{e1}$ between the yoke iron and the shell play a major role.

The process in which the torquer drive current $I_{T}$ generates voltage $V_{T}$ from the torquer coil can be expressed as:(10)$$H_{I_{T}}^{V_{T}}\left(j\omega_{I_{T}}\right)=\frac{V_{T}}{I_{T}}=R_{T}+R_{CS}+j\omega_{I_{T}}$$
where $\omega_{I_{T}}$ is the frequency of the torquer drive current $I_{T}$. When the accelerometer float shell is used, the pseudo-carrier signal $V_{p-sF}$ can be expressed as:(11)$$V_{p-sF}=\frac{C_{d2}}{C_{d2}+C_{s1}+C_{s2}}V_{T}=\frac{C_{d2}}{C_{d2}+2C_{0}}V_{T}$$

The pseudo-modulated signal $V_{m-sF}$ can be expressed as:(12)$$V_{m-sF}=\frac{2j\omega_{I_{T}}R_{f}}{1+2j\omega_{I_{T}}R_{f}C_{f}}\left(\frac{C_{d2}C_{s2}}{C_{d2}+C_{s2}}-\frac{C_{d2}C_{s1}}{C_{d2}+C_{s1}}\right)\frac{R_{2}}{R_{1}}V_{T}=\frac{2j\omega_{I_{T}}R_{f}}{1+2j\omega_{I_{T}}R_{f}C_{f}}\frac{2C_{d2}^{2}\Delta C}{{\left(C_{d2}+C_{0}\right)}^{2}-\Delta C^{2}}\frac{R_{2}}{R_{1}}V_{T}$$

In Equation (12), $\Delta C^{2}\ll {\left(C_{d2}+C_{0}\right)}^{2}$ can be ignored, and $V_{m-sF}$ can be expressed as:(13)$$V_{m-sF}=\frac{2j\omega_{I_{T}}R_{f}}{1+2j\omega_{I_{T}}R_{f}C_{f}}\frac{2C_{d2}^{2}}{{\left(C_{d2}+C_{0}\right)}^{2}}\frac{R_{2}}{R_{1}}V_{T}\Delta C$$

When the shell is grounded, the pseudo-carrier signal $V_{p-sG}$ can be expressed as:(14)$$V_{p-sG}=\frac{C_{d2}}{C_{d2}+C_{s1}+C_{s2}+C_{e1}}V_{T}=\frac{C_{d2}}{C_{d2}+2C_{0}+C_{e1}}V_{T}$$

The pseudo-modulated signal $V_{m-sG}$ can be expressed as:(15)$$V_{m-sG}=\frac{2j\omega_{I_{T}}R_{f}}{1+2j\omega_{I_{T}}R_{f}C_{f}}\left(\frac{C_{d2}C_{s2}}{C_{d2}+C_{s2}+C_{e1}}-\frac{C_{d2}C_{s1}}{C_{d2}+C_{s1}+C_{e1}}\right)\frac{R_{2}}{R_{1}}V_{T}\approx \frac{2j\omega_{I_{T}}R_{f}}{1+2j\omega_{I_{T}}R_{f}C_{f}}\frac{2C_{d2}^{2}}{{\left(C_{d2}+C_{e1}+C_{0}\right)}^{2}}\frac{R_{2}}{R_{1}}V_{T}\Delta C$$

The accelerometer closed-loop system is linear; $I_{T}$, $V_{T}$, and ΔC have the same frequency; and the maximum frequency is equal to the closed-loop system bandwidth. $V_{T}$ and ΔC multiply to produce a double-frequency effect, so that the frequency of $V_{m-s}$ is twice that of $I_{T}$, and then through the mixing step, the output-signal frequency is three times that of $I_{T}$. Generally, the cut-off frequency of the low-pass filter circuit of the carrier modulation detection circuit is several times that of the closed-loop system bandwidth; therefore, the attenuation of the frequency-doubling signal through the filter circuit can be ignored. The structure of the detection noise transfer system of the SCM detection circuit is as shown in Figure 4d. The detection noise $V_{n-s}$ of SCM detection circuit can be expressed as:(16)$$V_{n-s}=A_{Gain}K_{V_{T}}^{V_{p-s}}H_{V_{T}\cdot \Delta C}^{V_{m-s}}\left(2j\omega_{I_{T}}\right){\left(H_{I_{T}}^{V_{T}}\left(j\omega_{I_{T}}\right)\right)}^{2}K_{T}\theta \left(j\omega_{I_{T}}\right)K_{s}I_{T}^{3}$$
where $K_{V_{T}}^{V_{p-s}}$ represents the transfer process from $V_{T}$ to carrier signal $V_{p-s}$, $K_{V_{T}}^{V_{p-s}}=\frac{C_{d2}}{C_{d2}+2C_{0}}$ when the float shell is used, and $K_{V_{T}}^{V_{p-s}}=\frac{C_{d2}}{C_{d2}+2C_{0}+C_{e1}}$ when the shell is grounded; $H_{V_{T}\cdot \Delta C}^{V_{m-s}}\left(j\omega_{I_{T}}\right)$ represents the transfer process from the product of $V_{T}$ and ΔC to $V_{m-s}$. $H_{V_{T}\cdot \Delta C}^{V_{a-s}}\left(j\omega_{I_{T}}\right)=\frac{2j\omega_{I_{T}}R_{f}}{1+2j\omega_{I_{T}}R_{f}C_{f}}\frac{2C_{d2}^{2}}{{\left(C_{d2}+C_{0}\right)}^{2}}\frac{R_{2}}{R_{1}}$ when the float shell is used, and $H_{V_{T}\cdot \Delta C}^{V_{m-s}}\left(2j\omega_{I_{T}}\right)=\frac{2j\omega_{I_{T}}R_{f}}{1+2j\omega_{I_{T}}R_{f}C_{f}}\frac{2C_{d2}^{2}}{{\left(C_{d2}+C_{e1}+C_{0}\right)}^{2}}\frac{R_{2}}{R_{1}}$ when the shell is grounded.

### 3.2. Detection Noise Transfer System Structure of DCM Detection Circuit

When a DCM detection circuit is used to detect differential capacitance, the structure of the detection circuit is as shown in Figure 5a. $V_{s}$ and $-V_{s}$ are high-frequency sinusoidal carrier signals with the same frequency and amplitude and opposite phase, which are separately input to the two movable plates of the differential capacitor. After the output signals of the movable plate pass through the charge amplifier circuit, the modulated signal $V_{a}$ can be expressed as:(17)$$V_{a}=\frac{C_{s2}-C_{s1}}{\frac{1}{j\omega R_{f}}+C_{f}}V_{s}=\frac{2\Delta C}{\frac{1}{j\omega R_{f}}+C_{f}}V_{s}\approx \frac{2}{C_{f}}\Delta CV_{s}$$

As with the SCM detection circuit, after mixing, filtering, and gain adjustment of the modulated signal $V_{a}$ are performed, a voltage signal $V_{C}$ proportional to the change in differential capacitance ΔC can be obtained, and expressed as:(18)$$V_{C}=\frac{A_{Gain}A_{s}^{2}}{2C_{f}}\Delta C$$

The detection noise transfer path formed by the DCM detection circuit is shown in Figure 5b. After the voltage $V_{T}$ on the torquer coil passes through the distributed capacitance network, it is loaded onto the three plates of the differential capacitor, and then passed into the charge amplification circuit to form a pseudo-modulated signal $V_{m-d}$. The signal loaded on the top plate is also passed into the phase-shift circuit as a pseudo-carrier signal $V_{p-d}$. After mixing $V_{p-d}$ with $V_{m-d}$, the electric field coupling detection noise $V_{n-d}$ is formed through a low-pass filter and gain adjustment circuit. In the DCM detection noise transfer path, $C_{e1}$ is connected to the reverse port of the operational amplifier, which is approximately grounded and does not affect the differential capacitance detection. $C_{bb}$ does not divide the voltage or affect differential capacitance detection. We remove $C_{e1}$ and $C_{bb}$ to obtain a simplified detection noise transfer path of the DCM detection circuit, as shown in Figure 5c. In the transfer path of the detection noise of the DCM detection circuit, the distributed capacitances $C_{d1}$, $C_{d2}$, and $C_{d3}$ between the torquer coil and the three plates of the differential capacitor play a major role.

The pseudo-carrier signal $V_{p-d}$ can be expressed as
(19)$$V_{p-d}=\frac{C_{d3}}{C_{d3}+C_{s1}}V_{T}=\frac{C_{d3}}{C_{d3}+C_{0}-\Delta C}V_{T}\approx \frac{C_{d3}}{C_{d3}+C_{0}}V_{T}$$

The equivalent capacitance $C_{in}$ of the capacitance network composed of distributed capacitances $C_{d1}$, $C_{d2}$, and $C_{d3}$ and differential capacitors $C_{s1}$ and $C_{s2}$ can be expressed as
(20)$$C_{in}=\frac{C_{d3}C_{s1}}{C_{d3}+C_{s1}}+\frac{C_{d1}C_{s2}}{C_{d3}+C_{s2}}+C_{d2}=\frac{2C_{d3}\left(C_{0}^{2}-\Delta C^{2}\right)+2C_{d3}^{2}C_{0}}{{\left(C_{d3}+C_{0}\right)}^{2}-\Delta C^{2}}+C_{d2}\approx 2\frac{C_{d3}C_{0}}{C_{d3}+C_{0}}+C_{d2}\;$$

The pseudo-modulated signal $V_{m-d}$ can be expressed as
(21)$$V_{m-d}=-\frac{j\omega_{I_{T}}R_{f}C_{in}}{1+j\omega_{I_{T}}R_{f}C_{f}}V_{T}$$

The frequency of $V_{p-d}$ and $V_{m-d}$ is the same, and the frequency of the mixing output signal $V_{n-d}$ is twice that of $I_{T}$. The structure of the detection noise transfer system of the DCM detection circuit is shown in Figure 5d. The detection noise $V_{n-d}$ of the DCM detection circuit can be expressed as:(22)$$V_{n-d}=A_{Gain}K_{V_{T}}^{V_{p-d}}H_{V_{T}}^{V_{m-d}}\left(j\omega_{I_{T}}\right){\left(H_{I_{T}}^{V_{T}}\left(j\omega_{I_{T}}\right)\right)}^{2}I_{T}^{2}$$
where $K_{V_{T}}^{V_{p-d}}=\frac{C_{d3}}{C_{d3}+C_{0}}$ represents the transfer process from $V_{T}$ to carrier signal $V_{p-d}$, and $H_{V_{T}}^{V_{m-d}}\left(j\omega_{I_{T}}\right)=-\frac{j\omega_{I_{T}}R_{f}C_{in}}{1+j\omega_{I_{T}}R_{f}C_{f}}$ represents the transfer process from $V_{T}$ to $V_{m-d}$.

## 4. Analysis and Experiment in Relation to Detection Noise Characteristics

### 4.1. Analysis of Detection Noise Transfer in The Closed-Loop System of The QFA

The detection noise $V_{n}$ is superimposed onto the output signal $V_{C}$ of the differential capacitance detection circuit and enters the closed-loop system of the QFA. The introduction position of the detection noise $V_{n}$ is shown in Figure 6. The transfer function between $V_{n}$ and the drive current sampling output $V_{0}$ is
(23)$$H_{V_{n}}^{V_{0}}\left(s\right)=\frac{M\left(s\right)K_{I}K_{U}}{1-M\left(s\right)K_{I}K_{T}\theta \left(s\right)K_{s}K_{a}}$$

According to Equations (12), (21), and (22), when the SCM detection circuit is used to detect differential capacitance, the transfer relationship between the equivalent acceleration $a_{n-s}$ of the detected noise and input acceleration $a_{i}$ can be expressed as:(24)$$a_{n-s}=K_{V_{0}}^{a_{n}}H_{V_{n}}^{V_{0}}\left(\omega_{a_{i}}\right)V_{n-s}=K_{V_{0}}^{a_{n}}H_{V_{n}}^{V_{0}}\left(j\omega_{a_{i}}\right)H_{I_{T}^{3}}^{V_{n-s}}\left(j\omega_{a_{i}}\right){\left(H\left(j\omega_{a_{i}}\right)\right)}^{3}a_{i}^{3}$$
where $\omega_{a_{i}}$ is the frequency of input acceleration $a_{i}$, and $K_{V_{0}}^{a_{n}}$ is the transfer function between current sampling output $V_{0}$ and the equivalent acceleration $a_{n}$ of detection noise, equal to the reciprocal product of sampling circuit transfer function $K_{U}$ and accelerometer scale factor $K_{1}$. $H_{I_{T}^{3}}^{V_{n-s}}\left(j\omega_{a_{i}}\right)$ represents the transfer relationship between torquer converter drive current $I_{T}$ and detection noise $V_{n-s}$.

When a DCM detection circuit is used to detect differential capacitance, the transfer relationship between equivalent acceleration $a_{n-d}$ of the detected noise and input acceleration $a_{i}$ can be expressed as:(25)$$a_{n-d}=K_{V_{0}}^{a_{n}}H_{V_{n}}^{V_{0}}\left(\omega_{a}\right)V_{n-d}=K_{V_{0}}^{a_{n}}H_{V_{n}}^{V_{0}}\left(j\omega_{a}\right)H_{I_{T}^{2}}^{V_{n-d}}\left(j\omega_{a}\right){\left(H\left(j\omega_{a}\right)\right)}^{2}a_{i}^{2}$$
where $H_{I_{T}^{2}}^{V_{n-d}}\left(j\omega_{a_{i}}\right)$ represents the transfer relationship between the torquer drive current $I_{T}$ and detection noise $V_{n-d}$.

### 4.2. Analysis of Influencing Factors of Detection Noise

#### 4.2.1. Analysis of Influencing Factors of Detection Noise in SCM Detection Circuit

According to Equation (24), the magnitude of the detection noise equivalent acceleration $a_{n-s}$ of the SCM detection circuit is determined by H(s), $H_{V_{n}}^{V_{0}}\left(s\right)$, $H_{I_{T}^{3}}^{V_{n-s}}\left(s\right)$, and $K_{V_{0}}^{a_{n}}$. Under the condition that the structure of the accelerometer pendulum reed and the bandwidth of the closed-loop system are constant, $K_{B}$, θ(s), $K_{s}$, $K_{T}$, H(s), and $K_{V_{0}}^{a_{n}}$ are unchanged, and the product of $\theta \left(s\right)K_{s}K_{a}M\left(s\right)K_{I}K_{T}$ is fixed. If $K_{I}$ and $K_{a}$ are adjusted, M(s) will be adjusted in equal proportion. Adjusting $K_{I}$ only affects the transfer function $H_{V_{n}}^{V_{0}}\left(s\right)$, but $K_{I}$ in the numerator and denominator of $H_{V_{n}}^{V_{0}}\left(s\right)$ is multiplied by M(s); thus, adjusting $K_{I}$ does not change the value of the transfer function $H_{V_{n}}^{V_{0}}\left(s\right)$, and cannot suppress $a_{n-s}$.

According to Equation (9), the transfer function $K_{a}$ of the SCM detection circuit can be expressed as:(26)$$K_{a}=\frac{A_{diif}A_{Gain}A_{s}^{2}}{2C_{f}}$$

$K_{a}$ is proportional to $A_{diif}$, $A_{Gain}$, and $A_{S}$ squared, and inversely proportional to $C_{f}$. The changes in $A_{diif}$, $A_{Gain}$, and $C_{f}$ affect not only $K_{a}$ but also the transfer function $H_{I_{T}^{3}}^{V_{n-s}}\left(s\right)$. According to Equations (23), (24), and (26), when the accelerometer float shell is used, the equivalent acceleration of detection noise $a_{n-sF}$ is
(27)$$a_{n-sF}=K_{V_{0}}^{a_{n}}\frac{K_{a}M\left(s\right)K_{I}K_{U}K_{T}\theta \left(s\right)K_{s}}{1-M\left(s\right)K_{I}K_{T}\theta \left(s\right)K_{s}K_{a}}\frac{2sR_{f}}{\frac{1}{C_{f}}+2sR_{f}}\frac{2}{A_{s}^{2}}\frac{2C_{d2}^{2}}{{\left(C_{d2}+C_{0}\right)}^{2}}K_{V_{T}}^{V_{p-s}}{\left(H_{I_{T}}^{V_{T}}\left(s\right)\right)}^{2}{\left(H\left(s\right)\right)}^{3}a_{i}^{3}$$
when the shell is grounded, the equivalent acceleration of detection noise is
(28)$$a_{n-sG}=K_{V_{0}}^{a_{n}}\frac{K_{a}M\left(s\right)K_{I}K_{U}K_{T}\theta \left(s\right)K_{s}}{1-M\left(s\right)K_{I}K_{T}\theta \left(s\right)K_{s}K_{a}}\frac{2sR_{f}}{\frac{1}{C_{f}}+2sR_{f}}\frac{2}{A_{s}^{2}}\frac{2C_{d2}^{2}}{{\left(C_{d2}+C_{e1}+C_{0}\right)}^{2}}K_{V_{T}}^{V_{p-s}}{\left(H_{I_{T}}^{V_{T}}\left(s\right)\right)}^{2}{\left(H\left(s\right)\right)}^{3}a_{i}^{3}\;\;$$

According to Equations (27) and (28), increasing the amplitude of carrier signal $V_{s}$ and reducing the value of feedback capacitance $C_{f}$ can inhibit the detection of $a_{n-s}$.

#### 4.2.2. Analysis of Influencing Factors of Detection Noise in DCM Detection Circuit

According to Equation (24), the magnitude of the detection noise equivalent acceleration $a_{n-d}$ of the DCM detection circuit is determined by H(s), $H_{V_{n}}^{V_{0}}\left(s\right)$, $H_{I_{T}^{2}}^{V_{n-d}}\left(s\right)$, and $K_{V_{0}}^{a_{n}}$. As in the case of the single-carrier modulation detection circuit, adjusting $K_{I}$ cannot suppress $a_{n-d}$.

According to Equation (18), the transfer function $K_{a}$ of the DCM detection circuit can be expressed as:(29)$$K_{a}=\frac{A_{Gain}A_{s}^{2}}{2C_{f}}$$

According to Equations (23), (25), and (29), the equivalent acceleration of detection noise $a_{n-d}$ is
(30)$$a_{n-d}=-K_{V_{0}}^{a_{n}}\frac{K_{a}M\left(s\right)K_{I}K_{U}}{1-M\left(s\right)K_{I}K_{T}\theta \left(s\right)K_{s}K_{a}}K_{V_{T}}^{V_{p-d}}\frac{sR_{f}C_{in}}{\frac{1}{C_{f}}+sR_{f}}\frac{2}{A_{s}^{2}}{\left(H_{I_{T}}^{V_{T}}\left(s\right)\right)}^{2}{\left(H\left(s\right)\right)}^{2}a_{i}^{2}\;\;$$

Increasing the amplitude of carrier signal $V_{s}$ and reducing the value of feedback capacitance $C_{f}$ can inhibit the detection of $a_{n-d}$.

Through the analysis of the factors influencing the detection noise of the above two detection circuits, it can be seen that when the structure of the accelerometer pendulum reed and the bandwidth of the closed-loop system are constant, the factors influencing the detection noise of the two detection circuits are consistent, and the equivalent acceleration of the detection noise via electric field coupling can be suppressed by increasing the amplitude of the carrier signal of the detection circuit $V_{s}$ and reducing the value of the feedback capacitor $C_{f}$.

### 4.3. Experiment in Respect of Detection Noise Characteristics

In this study, the transfer characteristic and influencing factors of electric field coupling detection noise were analyzed through simulation. The torquer coil parameters of the typical QFA studied are $L_{T}=31.2\;mH$, $R_{T}=424\;\Omega$, $R_{CS}=1\;\Omega$. The typical parameters of the SCM and DCM detection circuits are $R_{1}=1\;K\Omega$, $R_{2}=1\;K\Omega$, $R_{f}=100\;M\Omega$, $C_{f}=20\;pF$ and $A_{Gain}=2.4$. The frequency of sinusoidal carrier signal $V_{s}$ is 50 kHz, the amplitude is 5 V and the typical value of $K_{a}$ is 1.5 V/pF. Typical parameters of the QFA closed-loop system in Figure 2 are as follows: $K_{B}=6.3\times 10^{-6}\;kg\cdot m$, $J=1.1\times 10^{-8}\;kg\cdot m^{2}$, $C=2.0\times 10^{-4}\;N\cdot m\cdot s/rad$, $K=2.2\times 10^{-3}\;N\cdot m/rad$, $K_{s}=12,000\;pF/rad$, $K_{T}=6.3\times 10^{-6}\;N\cdot m/A$, $K_{U}=1\;V/A$, and $K_{I}=10\;mA/V$. M(s) represents proportional feedback to meet the closed-loop bandwidth requirements, and the proportional feedback gain $K_{M}$ = 0.3. The accelerometer scale factor $K_{1}=1\;mA/g$, the acceleration measurement range is ±10 g, the system bandwidth is 300 Hz, and the effective resolution is 50 μg.

#### 4.3.1. Experiment on the Transfer Characteristics of Noise-Detection

The amplitude-frequency characteristics of H(s) and $H_{V_{n}}^{V_{0}}\left(s\right)$ are shown in Figure 7. The cut-off frequency of the closed-loop transfer function of the accelerometer is 300 Hz, which is consistent with the actual bandwidth of the system. $H_{V_{n}}^{V_{0}}\left(s\right)$ monotonically increases within the range of three times the input acceleration, and the transmission of detection noise in the closed-loop system is a high-pass characteristic.

When the system input acceleration is 1 g, the relationship between the detected noise equivalent acceleration $a_{n}$ and the input acceleration $a_{i}$ frequency is as shown in Figure 8. $a_{n-sF}$ is the detection noise equivalent acceleration of the SCM detection circuit when the float shell is used, $a_{n-sG}$ is the detection noise equivalent acceleration of the SCM detection circuit when the shell is grounded and $a_{n-d}$ is the detection noise equivalent acceleration of the DCM detection circuit. The magnitude of $a_{n}$ is positively correlated with the frequency of $a_{i}$. The higher the frequency of $a_{i}$, the greater the equivalent acceleration of detection noise. The transfer process from $a_{i}$ to $a_{n}$ shows a high-pass characteristic. The equivalent acceleration of detection noise of the SCM detection circuit reaches the maximum when the $a_{i}$ frequency is 70 Hz, and the equivalent acceleration of detection noise of the DCM detection circuit reaches the maximum when the $a_{i}$ frequency is 220 Hz.

The system inputs a random acceleration of 0~10 g and a frequency of 0~300 Hz, and the equivalent acceleration of detection noise of each type of carrier modulation detection circuit is shown in Figure 9. The mean of $a_{n-sF}$ is 10.3 μg, $a_{n-sG}$ is 0.2 μg, and $a_{n-d}$ is 41.7 μg. The equivalent acceleration of the detection noise is greater when the DCM detection circuit is used than when the SCM detection circuit is used. When an SCM detection circuit is used, the shell grounding can effectively reduce the equivalent acceleration of the detection noise.

The comparison of different types of noise values in the QFA system is shown in Table 7. The above analysis shows that the detection noise equivalent acceleration is close to the effective resolution of the accelerometer of 50 μg, which is a factor that cannot be ignored, as doing so affects the measurement accuracy of the QFA.

#### 4.3.2. Experiment on Influencing Factors of Detection Noise

In this section, the simulation analysis results for the factors affecting the detection noise are given in the case where the accelerometer pendulum structure and the closed-loop system bandwidth are constant. Within the allowable adjustment range of conventional carrier modulation detection circuit parameters, when $V_{s}$ and $C_{f}$ are adjusted, the change trend of $K_{a}$ and $K_{M}$ is as shown in Figure 10. $K_{a}$ increases with the increase in $V_{s}$ and decreases with the increase in $C_{f}$. The change trend of $K_{M}$ is the opposite. The product of $K_{a}$ and $K_{M}$ remains unchanged at 0.45, the transfer function of the accelerometer closed-loop system remains unchanged, and the system bandwidth remains unchanged at 300 Hz. Therefore, increasing the carrier signal $V_{s}$ magnitude and decreasing the feedback capacitance $C_{f}$ will not affect the closed-loop system bandwidth.

The influence of increasing $V_{s}$ on the amplitude-frequency characteristic curve of $a_{n}$ is shown in Figure 11. $V_{s}$ doubled, and the amplitude-frequency characteristic curve of $a_{n}$ decreases by 12 dB. At this time, the mean of $a_{n-sF}$ is 2.58 μg, $a_{n-sG}$ is 0.05 μg, and $a_{n-d}$ is 10.4 μg, while $a_{n}$ attenuates 12 dB. Increasing the magnitude of carrier signal $V_{s}$ can suppress the detection noise.

The influence of reducing $C_{f}$ on the amplitude-frequency characteristic curve of $a_{n}$ is shown in Figure 12. When the input acceleration frequency is less than 20 Hz, $C_{f}$ reduces by half, and the amplitude-frequency characteristic curve decreases by 6 dB. When the input acceleration frequency is greater than 20 Hz, the attenuation of the amplitude-frequency characteristic curve gradually decreases, and when it reaches more than 300 Hz, it is consistent with the original amplitude-frequency characteristic curve. When $C_{f}$ is reduced by one time, the mean of $a_{n-sF}$ is 8.48 μg, $a_{n-sG}$ is 0.2 ug, and $a_{n-d}$ is 35.3 μg, while the average attenuation of $a_{n}$ is 1.6 dB. Reducing the feedback capacitance $C_{f}$ of the detection circuit can suppress the detection noise. The effect of reducing $C_{f}$ on the low-frequency component of $a_{n}$ was better, but the effect on the high-frequency component was not obvious.

When $V_{s}$ and $C_{f}$ are adjusted, the change rule of the mean magnitude of $a_{n}$ is as shown in Figure 13. Increasing $V_{s}$ is more effective than decreasing $C_{f}$; when $V_{s}=6.55\;V$ and $C_{f}=4.8\;\mathrm{pF}$, the inhibition effect on $a_{n}$ is the same.

When $V_{s}$ is adjusted to the maximum magnitude and $C_{f}$ to the minimum value, the average magnitude of $a_{n}$ is as shown in Table 8. At the same time, when $V_{s}$ is a maximum and $C_{f}$ is a minimum, the suppression effect of the detection noise can be maximized.

The relationship between the average $a_{n}$ and $V_{s}$ can be approximately expressed as
(31)$$a_{n-avg}=k_{V}V_{s}^{-2}$$
where $a_{n-avg}$ is the average value of $a_{n}$, and $k_{V}$ is the gain coefficient. When the SCM detection circuit is adopted for the float shell, $k_{V}=2.6\times 10^{-4}$; when the SCM detection circuit is adopted for the grounded shell, $k_{V}=5.4\times 10^{-6}$; and when the DCM detection circuit is adopted, $k_{V}=1.1\times 10^{-3}$.

The relationship between the average $a_{n}$ and $C_{f}$ can be approximately expressed as
(32)$$a_{n-avg}=k_{f}\log\left(C_{f}\right)+k_{0}$$
where $k_{f}$ is the gain coefficient and $k_{0}$ is the bias coefficient. When the SCM detection circuit is adopted for the float shell, $k_{f}=7.0\times 10^{-6}$, $k_{0}=1.3\times 10^{-6}$; when the SCM detection circuit is adopted for the grounded shell, $k_{f}=1.5\times 10^{-7}$, $k_{0}=2.7\times 10^{-5}$; and when the DCM detection circuit is adopted, $k_{f}=2.7\times 10^{-5}$, $k_{0}=7.5\times 10^{-6}$.

When designing a QFA with $a_{n}$ accuracy higher than 1 μg, it is necessary to keep the average value of $a_{n}$ below 0.1 μg to ensure that it has a small impact on the QFA accuracy. However, according to the average value of $a_{n}$ in Table 5, it can be seen that within the allowable adjustment range of the conventional carrier modulation detection circuit parameters, the low noise requirement can be met only when the shell is grounded and the SCM detection circuit is adopted. In the other two cases, no matter how the circuit parameters are adjusted, $a_{n-avg}$ is always greater than 0.1 μg. According to Equations (31) and (32), in order to make $a_{n-avg}$ less than 0.1 μg, $V_{s}$ should be adjusted to above 51.0 V or $C_{f}$ should be adjusted to below 0.67 pF when the float shell and the detection circuit are adopted by SCM. When the DCM detection circuit is used, $V_{s}$ should be adjusted to above 104.9 V or $C_{f}$ should be adjusted to below 0.53 pF. According to Equations (9), (18), (13), and (21), increasing $V_{s}$ and $C_{f}$ improves the signal-to-noise ratio of the detection circuit by increasing the gain of the differential capacitance detection circuit. However, this method does not really reduce the detection noise of electric field coupling, which requires the gain of the differential capacitance detection circuit to be large enough, which is difficult to realize. Therefore, it is necessary to further reduce the detection noise of electric field coupling for a high-accuracy accelerometer.

## 5. Suppression Method of Electric Field Coupling Detection Noise

### 5.1. Optimization Analysis of Carrier Modulation Differential Capacitance Detection Circuit

According to the structure of the detection noise transfer system of the two carrier modulation detection circuits, the detection noise $V_{n}$ is equal to the product of $V_{m}$ and $V_{p}$. In order to truly reduce the detection noise, $V_{n}$, $V_{m}$, or $V_{p}$ should be reduced. Increasing the feedback capacitance $C_{f}$ can reduce $V_{m}$, but increasing $C_{f}$ will increase $a_{n-avg}$; therefore, $V_{n}$ can only be reduced by reducing $V_{p}$. Reducing $V_{p}$ can not only reduce the gain from the driving current to the detection-noise-transmission process but will also not cause an H(s), $H_{V_{n}}^{V_{0}}\left(s\right)$, and $K_{V_{0}}^{a_{n}}$ gain change. According to Figure 4b and Figure 5b, $V_{p}$ only passes through a phase-shifting circuit before mixing with $V_{m}$ through the multiplier. Since the carrier signal $V_{S}$ through the phase-shifting circuit is a high-frequency signal, and pseudo-carrier signal $V_{p}$ is a low-frequency signal within 300 Hz, the value of $V_{p}$ can be suppressed by adding a high-pass filter in front of the phase-shifting circuit. The optimized carrier-modulated differential capacitance detection circuit is shown in Figure 14.

The design of the second-order Butterworth high-pass filter is shown in Figure 15a. The cut-off frequency of the high-pass filter is 10 kHz, the gain is 0 dB, the stopband frequency is 300 Hz, and the stopband attenuation is −60 dB. The transfer function $H_{HPF}\left(s\right)$ can be expressed as
(33)$$H_{HPF}\left(s\right)=\frac{R_{1}R_{2}C_{1}C_{2}s^{2}}{1+\left(C_{1}+C_{2}\right)R_{2}s+R_{1}R_{2}C_{1}C_{2}s^{2}}\;\;\;\;\;\;\;\;\;\;\;=\frac{2.54\times 10^{-10}s^{2}}{1+2.26\times 10^{-5}s+2.54\times 10^{-10}s^{2}}$$

The amplitude-frequency characteristic curve of the high-pass filter in Figure 15b shows that the high-pass filter effectively suppresses the pseudo-carrier signal $V_{p}$, and the carrier signal $V_{s}$ is not affected.

The structure of the optimized electric field coupling detection noise transfer system is shown in Figure 16.

### 5.2. Analysis of Suppression Effect of Electric Field Coupling Detection Noise

When the system input acceleration is 1 g, the relationship between the detected noise equivalent acceleration $a_{n}$ and the input acceleration $a_{i}$ frequency before and after the optimization of the differential capacitance detection circuit is as shown in Figure 17. Compared with before optimization, $a_{n}$ decreases significantly after the addition of a high-pass filter, and attenuates 130 dB on average in the bandwidth of 300 Hz.

The system inputs a random acceleration of 0~10 g and a frequency of 0~300 Hz; the equivalent acceleration of detected noise after optimization is shown in Figure 18. The mean of $a_{n-sF}$ is 1.19 × 10^−3^ μg, $a_{n-sG}$ is 2.47 × 10^−5^ μg, and $a_{n-d}$ is 8.47 × 10^−3^ μg.

The comparison of the average value of the equivalent acceleration of the detection noise before and after the optimization of the differential capacitance detection circuit is shown in Table 9. After the detection circuit optimization, the average equivalent acceleration of the detection noise attenuates about 78 dB compared with that before the optimization, which can significantly suppress the electric field coupling detection noise and ensure that the $a_{n-avg}$ is less than 0.1 μg, effectively reducing the impact of the detection noise on the accuracy of the accelerometer.

## 6. Conclusions

This paper focuses on the analysis of the coupling mechanism of the internal electric field of the QFA and the influence of the electric field coupling detection noise on the accuracy of the QFA, establishing the equivalent circuit model of the internal electric field coupling detection noise of the accelerometer. The value of the distributed capacitance inside the accelerometer is simulated to obtain the detection noise transfer system structure of different carrier modulated differential capacitance detection circuits. Through the simulation experiment, the noise of electric field coupling detection is calculated. The experimental results show that the transmission of electric field coupling detection noise in the closed-loop system of the QFA is high-pass characteristic. Within the effective range and bandwidth of the system, the average value of the equivalent acceleration of noise detected by the SCM detection circuit when using the float shell is 10.3 μg; the average value of the equivalent acceleration of noise detected by the SCM detection circuit when the shell is grounded is 0.2 μg; and the average value of the equivalent acceleration of noise detected by the DCM detection circuit is 41.7 μg. The equivalent acceleration of the detection noise is close to the effective resolution of the accelerometer of 50 μg, indicating that the field coupling detection noise is a non-negligible factor affecting the measurement accuracy of the accelerometer. The analysis and experiment on the influence factors of electric field coupling detection noise show that when the structure of the accelerometer pendulum reed and the bandwidth of the closed-loop system are constant, increasing the magnitude of the carrier signal and reducing the feedback capacitance of the detection circuit can reduce the detection noise; however, the suppression degree is limited, and it is difficult to meet the noise requirements of the QFA with 1 μg accuracy. It is necessary to optimize the differential capacitance detection circuit to further reduce the noise of electric field coupling detection. Therefore, a high-pass filter is added at the front of the phase-shifting circuit, which attenuates the average value of the equivalent acceleration of the detection noise by about 78 dB, and the average value is less than 0.1 μg, effectively reducing the impact of the detection noise on the accuracy of the QFA. In the following research work, the structure of the QFA will be optimized to reduce the value of the distributed capacitance, suppress the electric field coupling detection noise from the source, and improve the performance of the QFA.

## Figures and Tables

**Figure 1 micromachines-14-00535-f001:**
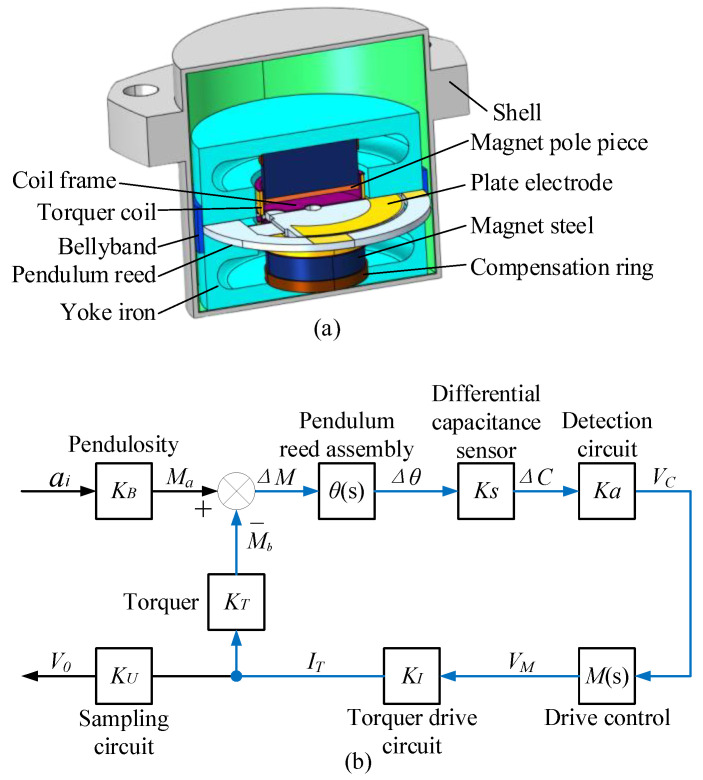
(**a**) Structure of QFA; (**b**) Closed-loop system structure of QFA.

**Figure 2 micromachines-14-00535-f002:**
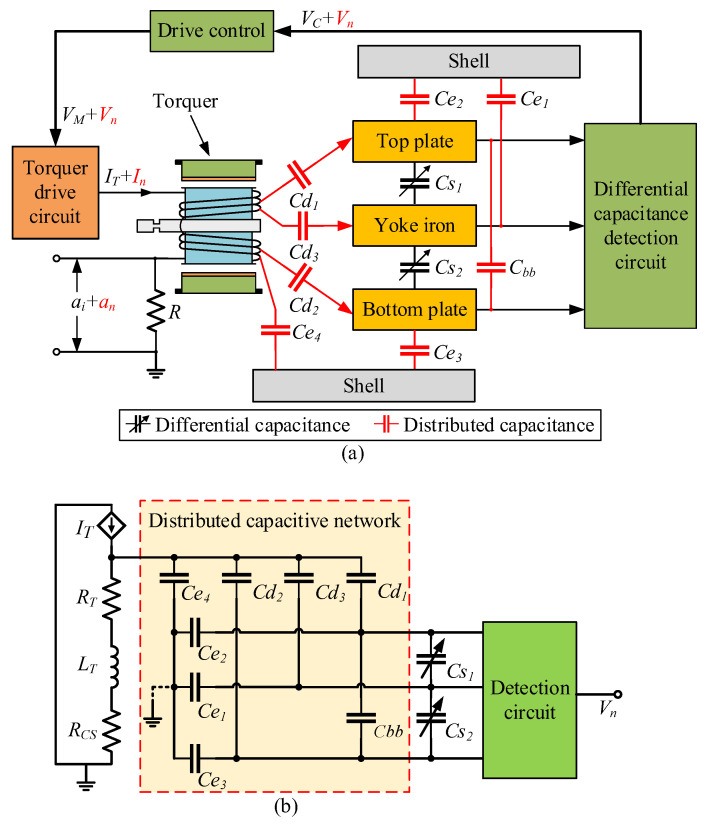
(**a**) Schematic diagram of distributed capacitance among various components of QFA; (**b**) Equivalent circuit model of electric field coupling detection noise in QFA.

**Figure 3 micromachines-14-00535-f003:**
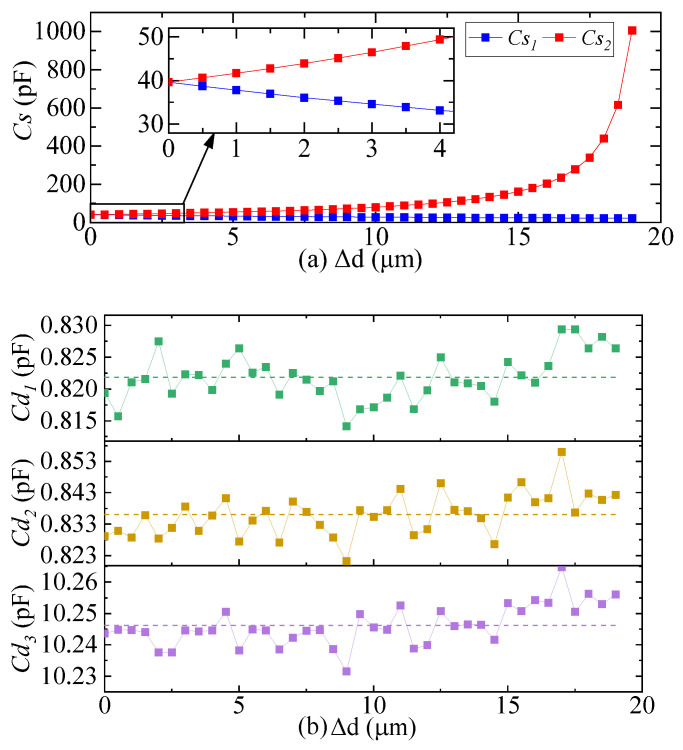
Simulation results of differential capacitance and distributed capacitance. (**a**) Simulation results of differential capacitance $C_{s1}$ and $C_{s2}$; (**b**) Simulation results of distributed capacitance $C_{d1}$, $C_{d2}$, and $C_{d3}$; (**c**) Simulation results of distributed capacitance $C_{e1}$, $C_{e2}$, $C_{e3}$, and $C_{e4}$; (**d**) Simulation results of distributed capacitance $C_{bb}$.

**Figure 4 micromachines-14-00535-f004:**
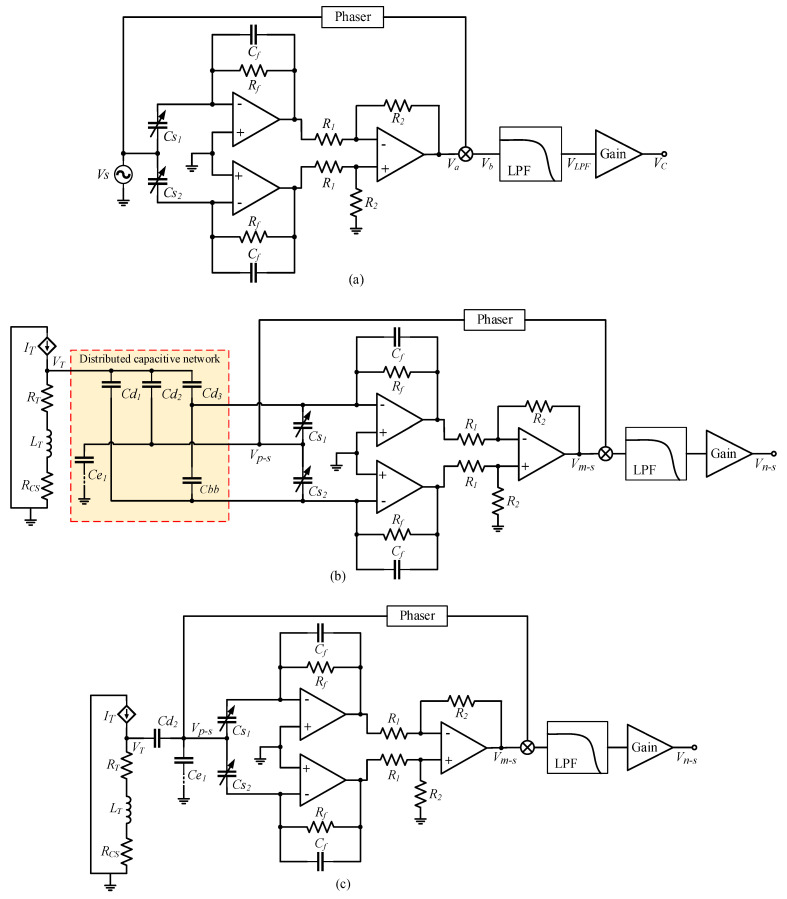
SCM detection circuit and its detection noise transfer path. (**a**) SCM detection circuit; (**b**) Detection noise transfer path composed of SCM detection circuit; (**c**) Simplified path of detection noise transfer formed by SCM detection circuit; (**d**) Structure of detection noise transfer system in SCM detection circuit.

**Figure 5 micromachines-14-00535-f005:**
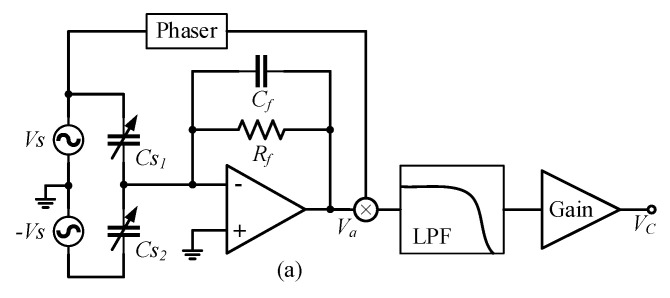
DCM detection circuit and its detection noise transfer path. (**a**) DCM detection circuit; (**b**) Detection noise transfer path composed of DCM detection circuit; (**c**) Simplified path of detection noise transfer formed by DCM detection circuit; (**d**) Structure of detection noise transfer system in DCM detection circuit.

**Figure 6 micromachines-14-00535-f006:**
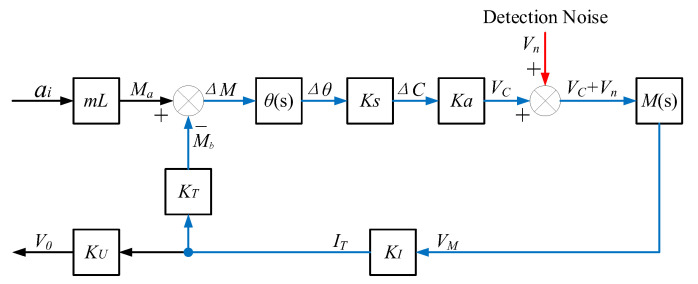
The introduction position of detection noise in the closed-loop system.

**Figure 7 micromachines-14-00535-f007:**
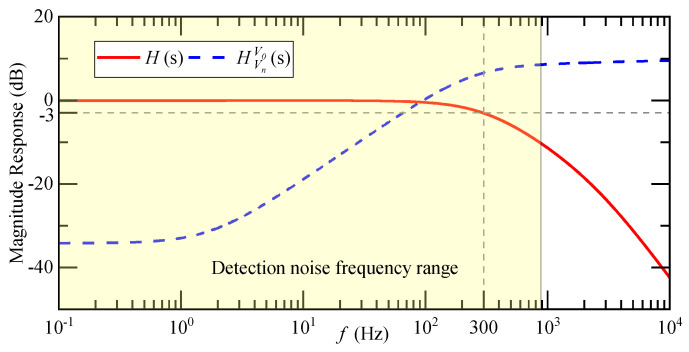
Amplitude-frequency characteristics of H(s) and $H_{V_{n}}^{V_{0}}\left(s\right)$.

**Figure 8 micromachines-14-00535-f008:**
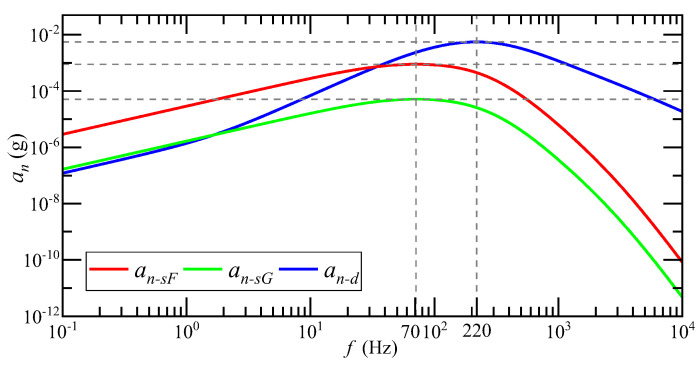
The relationship between the magnitude of $a_{n}$ and the frequency of $a_{i}$ under 1 g input acceleration.

**Figure 9 micromachines-14-00535-f009:**
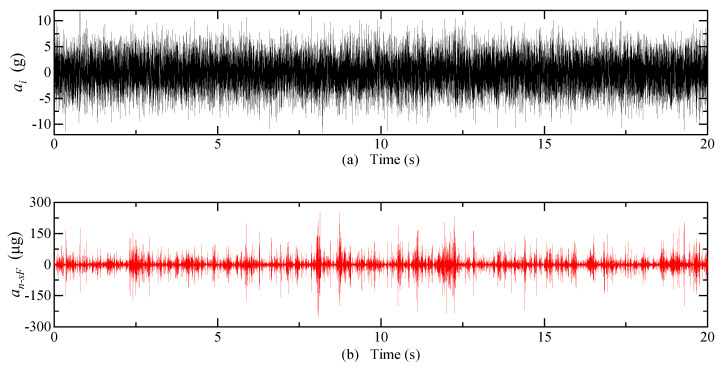
Electric field coupling detection noise simulation results. (**a**) Input acceleration; (**b**) The detection noise equivalent acceleration of the SCM detection circuit when the float shell is used; (**c**) The detection noise equivalent acceleration of the SCM detection circuit when the shell is grounded; (**d**) The detection noise equivalent acceleration of the DCM detection circuit.

**Figure 10 micromachines-14-00535-f010:**
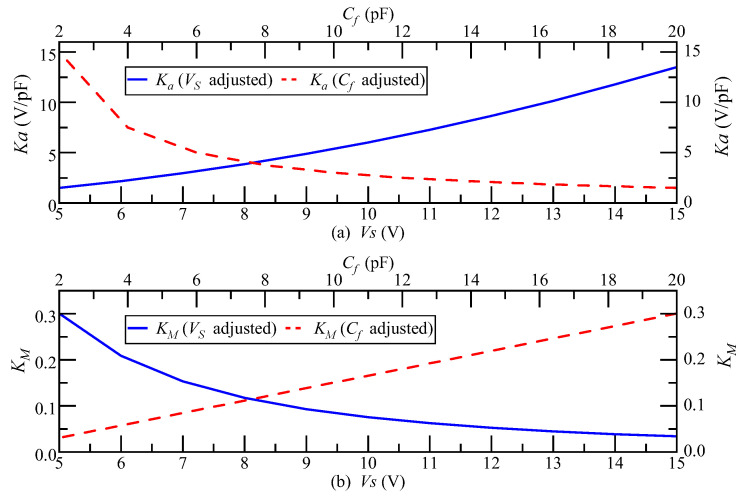
(**a**) The changing trend of $K_{a}$ when $V_{s}$ and $C_{f}$ are adjusted; (**b**) the changing trend of $K_{M}$ when $V_{s}$ and $C_{f}$ are adjusted.

**Figure 11 micromachines-14-00535-f011:**
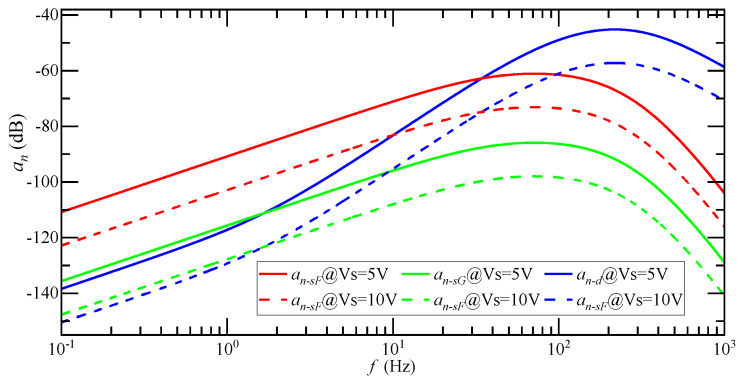
Amplitude-frequency characteristic curve of $a_{n}$ when $V_{s}$ is doubled.

**Figure 12 micromachines-14-00535-f012:**
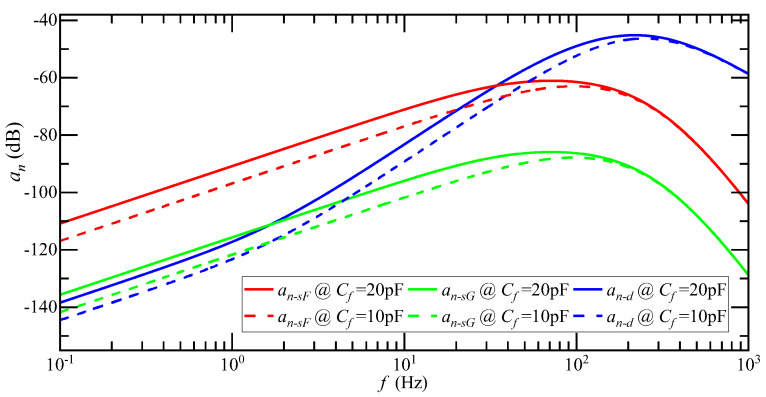
Amplitude-frequency characteristic curve of $a_{n}$ when $C_{f}$ decreases by one time.

**Figure 13 micromachines-14-00535-f013:**
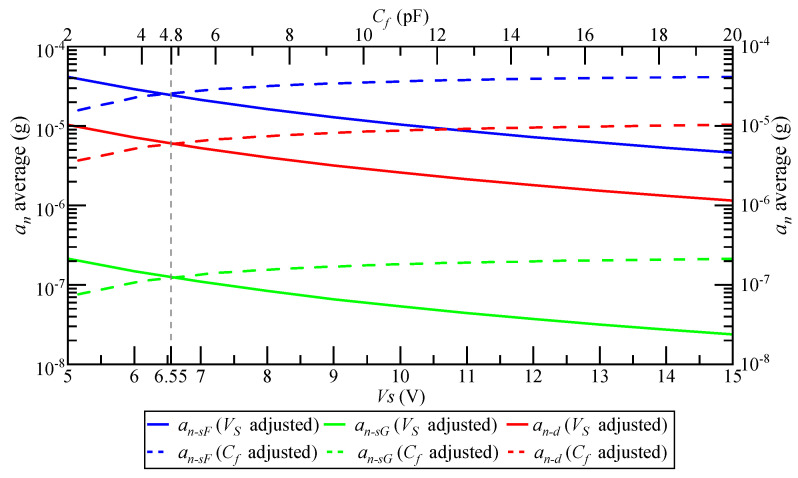
The change rule of the average magnitude of $a_{n}$ when $V_{s}$ and $C_{f}$ are adjusted.

**Figure 14 micromachines-14-00535-f014:**
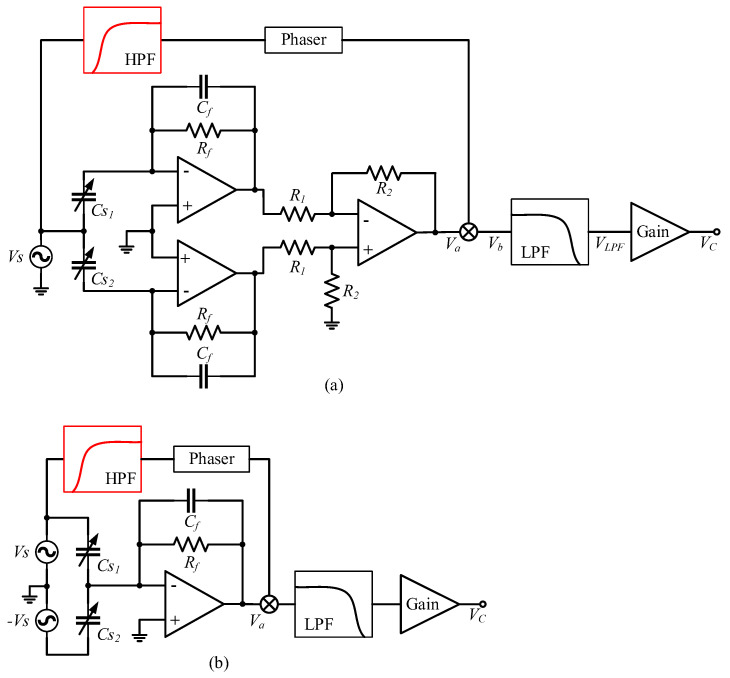
(**a**) The optimized SCM detection circuit; (**b**) The optimized DCM detection circuit.

**Figure 15 micromachines-14-00535-f015:**
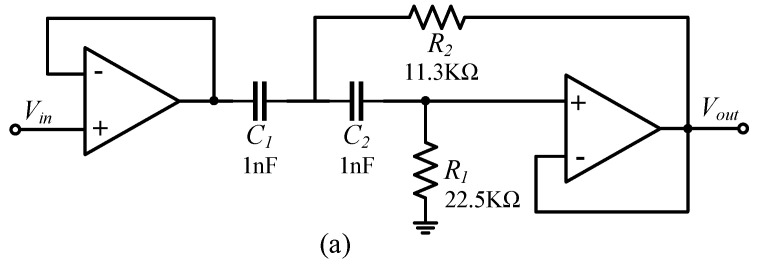
(**a**) High-pass filter structure; (**b**) Amplitude-frequency characteristic curve of high-pass filter.

**Figure 16 micromachines-14-00535-f016:**
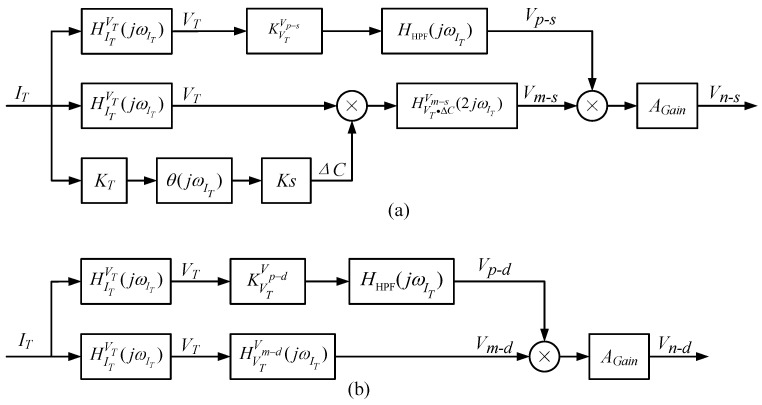
(**a**) Optimized SCM detection circuit noise transfer system structure; (**b**) Optimized DCM detection circuit noise transfer system structure.

**Figure 17 micromachines-14-00535-f017:**
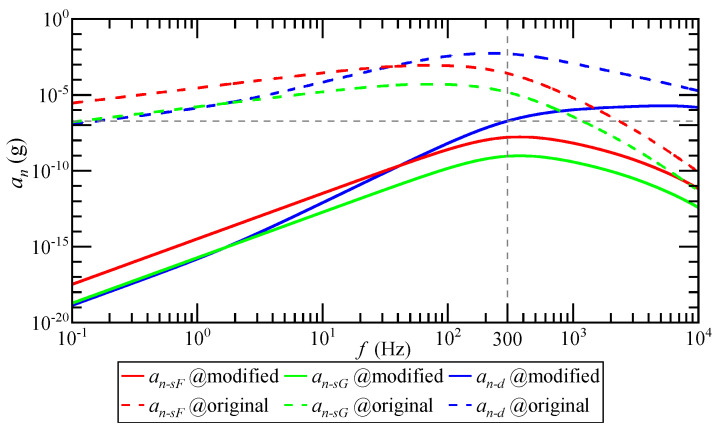
Before and after detection-circuit optimization, the relationship between the magnitude of $a_{n}$ and the frequency of $a_{i}$ under 1 g input acceleration.

**Figure 18 micromachines-14-00535-f018:**
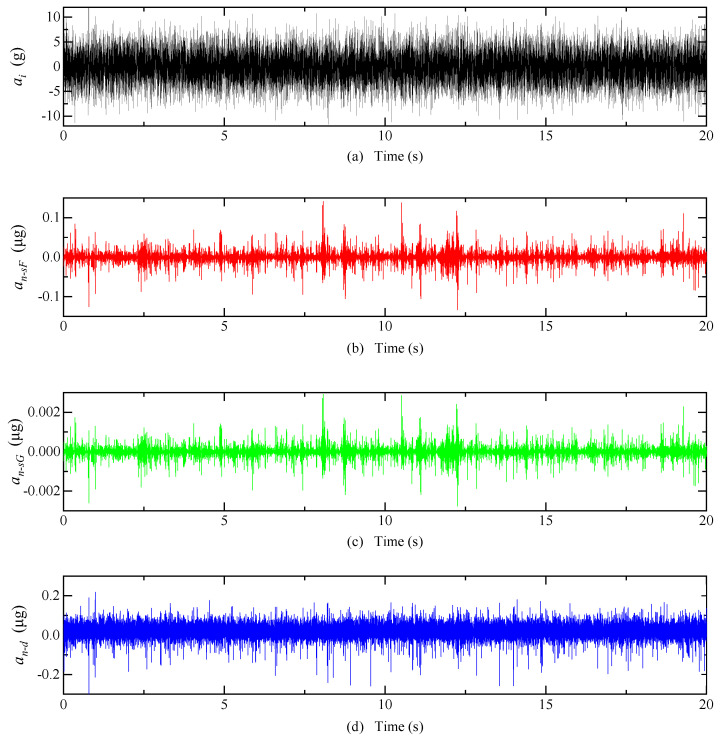
Simulation results of electric field coupling detection noise after optimization. (**a**) Input acceleration; (**b**) The detection noise equivalent acceleration of the SCM detection circuit when the float shell is used; (**c**) The detection noise equivalent acceleration of the SCM detection circuit when the shell is grounded; (**d**) The detection noise equivalent acceleration of the DCM detection circuit.

**Table 1 micromachines-14-00535-t001:** Main performance characteristics of QA3000-300, AI-Q-2010 and A600.

	QA3000-30	AI-Q-2010	A600
Input Range (g)	±60	±60	±60
Bias (mg)	4	4	4
Bias repeatability (μg)	40	550	20
Bias temperature sensitivity (μg/°C)	15	30	30
Scale Factor (mA/g)	1.20~1.46	1.20~1.46	1.20~1.46
Threshold (μg)	1	1	1
Bandwidth (Hz)	300	300	1000
Intrinsic Noise (μg RMS)	1500	1500	1500

**Table 2 micromachines-14-00535-t002:** Main performance characteristics of typical QFA.

Performance	Value
Input range (g)	±10
Bias (mg)	5
Bias repeatability (μg)	30
Bias temperature sensitivity (μg/°C)	30
Scale factor (mA/g)	1.00~1.20
Threshold (μg)	5
Bandwidth (Hz)	300
Intrinsic noise (μg RMS)	3000

**Table 3 micromachines-14-00535-t003:** Main geometric dimensions of QFA.

Size Name	Value (mm)	Size Name	Value (mm)
Shell height	25.00	Pendulum reed thickness	0.72
Shell diameter	38.20	Pendulum reed diameter	17.40
Meter head height	16.22	Coating area of pendulum reed	90.19 (mm^2^)
Meter head diameter	23.40	Length of flexible beam	2.80
Torquer coil height	2.40	Width of flexible beam	3.60
Torquer coil diameter	10.60	Thickness of flexible beam	0.02
Distance between upper and lower yoke iron	0.02	Flexible beam spacing	2.50

**Table 4 micromachines-14-00535-t004:** Material parameters of components of QFA.

Part Name	Material Type	Relative Permittivity
Shell	1Cr18Ni9Ti	1.00
Pendulum reed	JGS1	3.83
Coating film	Au	1.00
Torquer coil	Cu	1.00
Coil frame	Al2O3	9.50
Magnet steel	XGS240/46	1.00
Magnet pole piece	1J50	1.00
Compensation ring	1J38	1.00
Yoke iron	4J36	1.00
Bellyband	4J36	1.00
Adhesive tape	3M8992	3.10
Underfill	DG-3S	2.70
Filling gas	Air	1.00

**Table 5 micromachines-14-00535-t005:** Excitation voltage value of each component.

Part Name	Static Pole Plate	Torquer Coil	Top Plate	Bottom Plate	Shell
Voltage (V)	0.00	6.00	5.00	−5.00	0.00

**Table 6 micromachines-14-00535-t006:** Distributed capacitance value.

Capacitance Label	$C_{d1}$	$C_{d2}$	$C_{d3}$	$C_{bb}$	$C_{e1}$	$C_{0}$
Value (pF)	0.8	10.0	0.8	4.0	159.0	40.0

**Table 7 micromachines-14-00535-t007:** Comparison of different types of noise values in QFA system.

Noise Type	Value (μg)
Mechanical thermal noise	1.3 × 10^−5^
Bias repeatability	30
Detect circuit noise	1.53
Electric field coupling detection noise	41.7

**Table 8 micromachines-14-00535-t008:** Average value of $a_{n}$ for different values of $V_{s}$ and $C_{f}$.

$V_{S}(V)$	$C_{f}\;(\mathrm{pF})$	$a_{n-sF}\mathrm{Average}\;(\mu g)$	$a_{n-sG}\mathrm{Average}\;(\mu g)$	$a_{n-d}\mathrm{Average}\;(\mu g)$
15.0	20.0	1.1	0.024	4.6
5.0	2.0	3.4	0.071	14.8
15.0	2.0	0.4	0.0079	1.6

**Table 9 micromachines-14-00535-t009:** The average value of the detection noise equivalent acceleration before and after the optimization of the differential capacitor detection circuit.

Equivalent Acceleration of Detection Noise	Value Before Optimization (μg)	Optimized Value (μg)	Attenuation (dB)
$a_{n-sF}$ average	10.3	1.19 × 10^−3^	78.7
$a_{n-sG}$ average	0.2	2.47 × 10^−5^	78.2
$a_{n-d}$ average	41.7	8.47 × 10^−3^	73.8

## Data Availability

Not applicable.

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
