# Peer review of "Analysis of Noise-Detection Characteristics of Electric Field Coupling in Quartz Flexible Accelerometer"

_micromachines, 2023, doi:10.3390/mi14030535_

Round 1

Reviewer 1 Report (Previous Reviewer 2)

Comparisons of sensors are needed, especially the noise of the sensor. For example, the simulation results should compare with the experiments. 

Author Response

Dear reviewer:

Thank you for the reviewers’ comments on our manuscript entitled " Analysis of Noise-detection Characteristics of Electric Field Coupling in Quartz Flexible Accelerometer " (manuscript No. 2224006). Those comments are very helpful for revising and improving our paper, as well as the important guiding significance to other research. We have studied the comments carefully and made corrections which we hope meet with approval. The main corrections are in the manuscript and the responds to the reviewers’ comments are as follows.

Point 1: Comparisons of sensors are needed, especially the noise of the sensor. For example, the simulation results should compare with the experiments.

Response 1: According to your suggestion, we have added Table 6 in the analysis of noise suppression effect of electric field coupling detection (Section 5.2), comparing the average value of equivalent acceleration of noise detection before and after the optimization of detection circuit, so that readers can more intuitively understand the noise suppression effect of electric field coupling detection. Table 6 is as follows :

Table 6. The average value of the noise equivalent acceleration before and after the optimization of the differential capacitor detection circuit.

Equivalent Acceleration

of Detected Noise

Value Before

Optimization (μg)

Optimized

Value (μg)

Attenuation (dB)

an-sF average

10.3

1.19×10-3

78.7

 an-sG average

0.2

2.47×10-5

78.2

 an-d average

41.7

8.47×10-3

73.8

We are sorry that your suggestion of comparing simulation and experimental results with our existing accelerometer products and experimental equipment is temporarily unable to meet the conditions for actual experiments, mainly for the following reasons:

  1. There are many factors that affect the accuracy of the accelerometer. We have only studied the noise related to the detection and driving circuit. We cannot change the measurement error caused by the inherent structural factors such as accelerometer structure, pendulum structure, flexible beam thickness, etc.. However, the influence of these noise on the accelerometer is also important, and there is a correlation between these factors and the detection and drive circuit. Therefore, it is impossible to prove whether the precision reading errors detected by actual experiments are caused by a single noise source, electric field coupled noise.
  2. In order to measure the acceleration variation of less than 1μg, experimental measurements are required in an underground laboratory with minimal vibration. At the present stage, we do not have such experimental environment and experimental conditions, so we can only analyze the value of electric field coupling noise and the effect of noise suppression method through simulation experiment.

In the review of the last version of the paper, another reviewer also put forward the same suggestion as you, and we also explained the reason to him and got recognition.

We hope that you can understand the difficulties we have encountered at this stage, and we will continue to improve this experimental work when conditions are met.

Once again, thank you very much for your constructive comments and suggestions which would help us both in English and in depth to improve the quality of the paper.

Sincerely yours,

Zhigang Zhang and Lijun Tang on behalf of the authors.

Corresponding author: Lijun Tang at School of Physics and Electronic Science, Changsha University of Science and Technology, 410114, China, Changsha, [email protected], phone number: +86-139-7438-3567

Reviewer 2 Report (Previous Reviewer 1)

Compared with the previous version, this manuscript uses clearer pictures and more concise presentation. I think it can be published, but some issues still need to be corrected.

1.      Ref. 25 didn’t appear but still remains in the reference list.

2.      I suggest extending the conclusion part like the previous version to clarify the research.

Author Response

Dear reviewer:

Thank you for the reviewers’ comments on our manuscript entitled " Analysis of Noise-detection Characteristics of Electric Field Coupling in Quartz Flexible Accelerometer " (manuscript No. 2224006). Those comments are very helpful for revising and improving our paper, as well as the important guiding significance to other research. We have studied the comments carefully and made corrections which we hope meet with approval. The main corrections are in the manuscript and the responds to the reviewers’ comments are as follows.

Point 1: Ref. 25 didn’t appear but still remains in the reference list.

Response 1: Thank you for your careful review. We have checked the submitted manuscript of the paper and found no Ref. 25 in the manuscript. This error should be a minor mistake made by the editors of the journal after the second editing of the paper, which has been deleted.

Point 2: I suggest extending the conclusion part like the previous version to clarify the research.

Response 2: According to your valuable suggestions, we have modified the conclusion section of the paper. The revised conclusion section includes three parts. Firstly, the main work content and experimental results of the research on noise characteristics of quartz flexible accelerometer are summarized. Then, the suppression measures and effects of electric field coupling detection noise proposed in this paper are introduced. Finally, the follow-up research plan is prospected. Through the content of these three aspects, the conclusion is more substantial and more clearly describes the research work and experimental results of this paper. The conclusion section is as follows:

6. Conclusions

This paper focuses on the analysis of the coupling mechanism of the internal electric field of the QFA and the influence of the electric field coupling detection noise on the accuracy of the QFA, establishing the equivalent circuit model of the internal electric field coupling detection noise of the accelerometer. The value of the distributed capacitance in-side the accelerometer is simulated to obtain the detection noise transfer system structure of different carrier modulated differential capacitance detection circuits. Through the simulation experiment, the noise of electric field coupling detection is calculated. The experimental results show that the transmission of electric field coupling detection noise in the closed-loop system of the QFA is high-pass characteristic. Within the effective range and bandwidth of the system, the average value of the equivalent acceleration of noise detected by the SCM detection circuit when the float shell is 10.3 μg; the average value of the equivalent acceleration of noise detected by the SCM detection circuit when the shell is ground-ed is 0.2 μg; and the average value of the equivalent acceleration of noise detected by the DCM detection circuit is 41.7 μg. The equivalent acceleration of the detection noise is close to the effective resolution of the accelerometer of 50 μg, indicating that the field coupling detection noise is a non-negligible factor affecting the measurement accuracy of the accelerometer. The analysis and experiment on the influence factors of electric field coupling detection noise show that when the structure of the accelerometer pendulum reed and the bandwidth of the closed-loop system are constant, increasing the magnitude of the carrier signal and reducing the feedback capacitance of the detection circuit can reduce the detection noise; but the suppression degree is limited, and it is difficult to meet the noise requirements of the QFA with 1 μg accuracy. It is necessary to optimize the differential capacitance detection circuit to further reduce the noise of electric field coupling detection. Therefore, a high-pass filter is added at the front of the phase-shifting circuit, which attenuates the average value of the equivalent acceleration of the detection noise by about 78 dB, and the average value is less than 0.1 μg, effectively reducing the impact of the detection noise on the accuracy of the QFA. In the following research work, the structure of the QFA will be optimized to reduce the value of the distributed capacitance, suppress the electric field coupling detection noise from the source, and improve the performance of the QFA.

Once again, thank you very much for your constructive comments and suggestions which would help us both in English and in depth to improve the quality of the paper.

Sincerely yours,

Zhigang Zhang and Lijun Tang on behalf of the authors.

Corresponding author: Lijun Tang at School of Physics and Electronic Science, Changsha University of Science and Technology, 410114, China, Changsha, [email protected], phone number: +86-139-7438-3567

This manuscript is a resubmission of an earlier submission. The following is a list of the peer review reports and author responses from that submission.

Round 1

Reviewer 1 Report

This paper emphasizes the importance of electric field coupling in the accuracy of the accelerometer through a series of modeling and simulations. However, the argument of the article is not sufficient, and major revision is needed to increase the persuasiveness.

1.     Please check the English writing carefully. Here are some typical examples:

In line 39, there may be a typo “I” in the first sentence.

In line 43, it should be “a simpler structure”.

In line 49, it should be “model of the circuit”, also same to line 50.

2.     In the introduction section, the authors give out many previous researches about the optimization, but few evidences about the significance of the study on the internal electric field coupling noise. I suggest the authors to include more information about it to enhance the significance of this paper.

3.     The author evaluates the importance of detecting electric field coupling effect via a series of modelling and simulations. I suggest the authors performing some practical experiments to assist the results.

4.     In figure 3, there are some hand-drawn circuits on the schematic. However, as a reader, the plotting method confuses me a lot. Is there any better way to describe the distributed capacitances?

5.     In the conclusion section, the only derived result is that the electric field coupling detecting noise is a non-negligible factor. This conclusion is too simple to make an objective summary of the article. I suggest adding more discussions, such as giving suggestions on the degree of influence, optimization direction, etc., to increase the persuasiveness of the paper.

Reviewer 2 Report

This paper analyzed the noise characteristics of Quartz Flexible Accelerometer. This paper just described where the noise source come and how the noise effect on the QFA without further solution. There are no new methods, thoughts and principles.